# CONTRASTIVE VALUE LEARNING: IMPLICIT MODELS FOR SIMPLE OFFLINE RL

## ABSTRACT

Model-based reinforcement learning (RL) methods are appealing in the offline setting because they allow an agent to reason about the consequences of actions without interacting with the environment. Prior methods learn a 1-step dynamics model, which predicts the next state given the current state and action. These models do not immediately tell the agent which actions to take, but must be integrated into a larger RL framework. Can we model the environment dynamics in a different way, such that the learned model does directly indicate the value of each action? In this paper, we propose Contrastive Value Learning (CVL), which learns an implicit, multi-step model of the environment dynamics. This model can be learned without access to reward functions, but nonetheless can be used to directly estimate the value of each action, without requiring any TD learning. Because this model represents the multi-step transitions implicitly, it avoids having to predict high-dimensional observations and thus scales to high-dimensional tasks. Our experiments demonstrate that CVL outperforms prior offline RL methods on complex continuous control benchmarks.

## 1 INTRODUCTION

While the offline RL setting is relevant to many real-world applications where the ability for online data collection is limited, it often requires RL algorithms to find policies that are not well-supported by the training data. Instead of learning via trial-and-error, offline RL algorithms must leverage logged historical data to learn about the outcome of different actions, potentially by capturing environment dynamics as a proxy signal. Many prior approaches for this offline RL setting have been proposed, whether in model-free (Wu et al., 2019; Fujimoto et al., 2019; Kumar et al., 2020) or model-based (Kidambi et al., 2020; Yu et al., 2021) settings. Our focus will be on those that address this prediction problem head-on: by learning a predictive model of the environment which can be used in conjunction with most model-free algorithms.

Prior model-based methods (Yu et al., 2020b; Argenson and Dulac-Arnold, 2020; Kidambi et al., 2020; Yu et al., 2021) learn a model that predicts the observation at the next time step. This model is then used to generate synthetic data that can be passed to an off-the-shelf RL algorithm. While these approaches can work well on some benchmarks, they can be complex and expensive: the model must predict high-dimensional observations, and determining the value of an action may require unrolling the model for many steps. Learning a model of the environment has not made the RL problem any simpler. Moreover, as we will show later in the paper, the environment dynamics are intertwined with the policy inside the value function; model-based methods aim to decouple these quantities by separately estimating them. On the other hand, we show that one can directly learn a long-horizon transition model for a given policy, which is then used to estimate the value function. A natural use case for learning this long-horizon transition model (specifically, a state occupancy measure) from unlabelled data is multi-task pretraining, where the implicit dynamics model is trained on trajectory data across a collection of tasks, often exhibiting positive transfer properties. As we demonstrate in our experiments, this multi-task occupancy measure can then be finetuned using reward-labelled states on the task of interest, greatly improving performance upon existing pretraining methods as well as *tabula rasa* approaches.

In this paper, we propose to learn a different type of model for offline RL, a model which *(1)* will not require predicting high-dimensional observations and *(2)* can be directly used to estimate Q-values without requiring either model-based rollouts or model-free temporal difference learning. Precisely, we will learn an implicit model of the discounted state occupancy measure, i.e. a function which takes in a state, action and future state and outputs a scalar proportional to the likelihood of visiting the future state under some fixed policy.

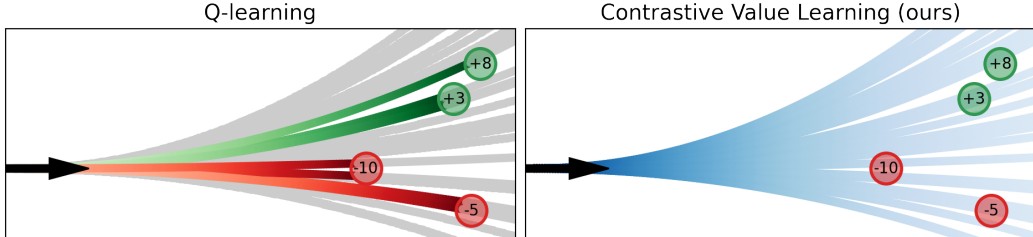

Figure 1: **Contrastive Value Learning**: A stylized illustration of trajectories (grey) and the rewards at future states (e.g., +8, -5). *(Left)* Q-learning estimates Q-values by "backing up" the rewards at future states. *(Right)* Our method learns the Q-values by fitting an implicit model to estimate the likelihoods of future states (blue), and taking the reward-weighted average of these likelihoods.

We will learn this implicit model via contrastive learning, treating it as a classifier rather than a generative model of observations. Once learned, we predict the likelihood of reaching every reward-labeled state. By weighting these predictions by the corresponding rewards, we form an unbiased estimate of the Q-function. Whereas methods like Q-learning estimate the Q-function of a state "backing up" reward values, our approach goes in the opposite direction, "propagating forward" predictions about where the agent will go.

We name our proposed algorithm Contrastive Value Learning(CVL). CVL is a simple algorithm for offline RL which learns the future state occupancy measure using contrastive learning and re-weights it with the future reward samples to construct a quantity proportional to the true value function. Because CVL represents multi-step transitions implicitly, it avoids having to predict high-dimensional observations and thus scales to high-dimensional tasks. Using the same algorithm, we can handle settings where reward-free data is provided, which cannot be directly handled by classical offline RL methods such as FQI (Munos, 2003) or BCQ (Fujimoto et al., 2019). We compare our proposed method to competitive offline RL baselines, notably CQL (Kumar et al., 2020) and CQL+UDS (Yu et al., 2022) on an offline version of the multi-task Metaworld benchmark (Yu et al., 2020a), and find that CVL greatly outperforms the baseline approaches as measured by the `rliable` library (Agarwal et al., 2021b). Additional experiments on image-based tasks from this same benchmark show that our approach scales to high-dimension tasks more seamlessly than the baselines. We also conduct a series of ablation experiments highlighting critical components of our method.

## 2 RELATED WORKS

Prior work has given rise to multiple offline RL algorithms, which often rely on behavior regularization in order to be well-supported by the training data. The key idea of offline RL methods is to balance interpolation and extrapolation errors, while ensuring proper diversity of out-of-dataset actions. Popular offline RL algorithms such as BCQ and CQL rely on a behavior regularization loss (Wu et al., 2019) as a way to control the extrapolation error. This regularization term ensures that the learned policy is well-supported by the data, i.e. does not stray too far away from the logging policy. The major issue with current offline RL algorithms is that they fail to fully capture the entire distribution over state-action pairs present in the training data.

To directly learn a value function using policy or value iteration, one needs to have information about the transition model in the form of sequences of state-action pairs, as well as the reward emitted by this transition. However, in some real-world scenarios, the reward might only be available for a small subset of data. For instance, in the case of recommending products available in an online catalog to the user, the true long-term reward (user buys the product) is only available for users who have browsed the item list for long enough and have purchased a given item. It is possible to decompose the value function into reward-dependent and reward-free parts, as was done by (Barreto et al., 2016) through the successor representation framework (Dayan, 1993). More recent approaches (Janner et al., 2020; Eysenbach et al., 2020; 2022) use a generative model to learn the occupancy measure over future states for each state-action pair in the dataset; its expectation corresponds to the successor representation. However, learning an explicit multi-step model such as (Janner et al., 2020) can be unstable due to the bootstrapping term in the temporal difference loss. Similarly to model-based approaches, our method will learn a reward-free representation of the world, but will do so without having to predict high-dimensional observations and without having to do costly autoregressive rollouts. Thus, while our critic is trained without requiring rewards, it is much more similar to a value function than a standard 1-step model.

Learning a conditional probability distribution over a highly complex space can be a challenging task, which is why it is often easier to instead approximate it using a density ratio specified by an inner product in a much lower-dimensional latent space. To learn an occupancy measure over future states without passing via the temporal difference route, one can use noise-contrastive estimation (NCE, Gutmann and Hyvärinen, 2010; Oord et al., 2018) to approximate the corresponding log ratio of densities as an implicit function. Contrastive learning was originally proposed as an alternative to classical maximum likelihood estimation, but has since then seen successes in static self-supervised learning (He et al., 2020; Chen et al., 2020). In reinforcement learning, NCE was shown to improve the robustness of state representations to exogenous noise (Srinivas et al., 2020; Mazoure et al., 2020; Agarwal et al., 2021a) and, more recently, to be an efficient replacement for traditional goal-conditioned methods (Eysenbach et al., 2022).

## 3 PRELIMINARIES

**Reinforcement learning** We assume a Markov decision process $M$ defined by the tuple $M = \langle \mathcal{S}, S_0, \mathcal{A}, \mathcal{T}, r, \gamma \rangle$, where $\mathcal{S}$ is a state space, $S_0 \subseteq \mathcal{S}$ is the set of starting states, $\mathcal{A}$ is an action space, $\mathcal{T} = \mathbb{P}[\cdot | s_t, a_t] : \mathcal{S} \times \mathcal{A} \to \Delta(\mathcal{S})$ is a one-step transition function[1], $r : \mathcal{S} \times \mathcal{A} \to [r_{\min}, r_{\max}]$ is a reward function and $\gamma \in [0,1)$ is a discount factor. The system starts in one of the initial states $s_0 \in S_0$. At every timestep $t = 1,2,3,..$, the policy $\pi : \mathcal{S} \to \Delta(\mathcal{A})$, samples an action $a_t \sim \pi(\cdot | o_t)$. The environment transitions into a next state $s_{t+1} \sim \mathcal{T}(\cdot | s_t, a_t)$ and emits a reward $r_t = r(s_t, a_t)$. The aim is to learn a Markovian policy $\pi(a | s)$ that maximizes the discounted sum of returns over an episode of length $H$:

$$\max_{\pi \in \Pi} \mathbb{E}_{\mathbb{P}_{0:H}^\pi, S_0} \left[ \sum_{t=0}^{H} \gamma^t r(s_t, a_t) \right], \tag{1}$$

where $\mathbb{P}_{t:t+K}^\pi$ denotes the joint distribution of $\{s_{t+k}, a_{t+k}\}_{k=1}^K$ obtained by executing $\pi$ in the environment for $K$ timesteps starting at timestep $t$.

We assume that the offline RL algorithm cannot interact with the environment, but instead must learn from an offline dataset of logged trajectories $\{(s_0, a_0, s_1, a_1, \cdots)\}$. Value-based RL algorithms maximize cumulative episodic rewards by estimating the state-action value function under a policy $\pi$:

$$Q^\pi(s_t, a_t) = \mathbb{E}_{\mathbb{P}_t^\pi} \left[ \sum_{k=1}^{H} \gamma^k r(s_{t+k}, a_{t+k}) | s_t, a_t \right], \tag{2}$$

for $s_t \in \mathcal{S}, a_t \in \mathcal{A}$. Alternatively, the value function can be written as the expectation of the reward over the discounted occupancy measure:

$$Q^\pi(s_t, a_t) = \frac{1}{1-\gamma} \mathbb{E}_{s,a \sim \mathbb{P}_{t:H}^\pi(s_t, a_t), \pi(s)} [r(s,a)], \tag{3}$$

where $\mathbb{P}_{t:H}^\pi(s | s_t, a_t) = (1-\gamma) \sum_{\Delta t=1}^{H} \gamma^{\Delta t - 1} \mathbb{P}[S_{t+\Delta t} = s | s_t, a_t; \pi]$ as defined in Janner et al. (2020). Note that the occupancy measure can equivalently be re-written in terms of the geometric distribution over the time interval $[0, \infty)$ for infinite-horizon rollouts:

$$\mathbb{P}_{0:\infty}^\pi(s | s_0, a_0) = \mathbb{E}_{\Delta t \sim \text{Geom}(1-\gamma)} [\mathbb{P}[S_{t+\Delta t} | s_0, a_0; \Delta t; \pi]] \tag{4}$$

This decomposition of the value function has already been used in previous works based on the successor representation (Dayan, 1993; Barreto et al., 2016) and, more recently, $\gamma$-models (Janner et al., 2020). We will use it to efficiently learn an implicit density ratio proportional to the state occupancy measure using contrastive learning.

**Noise-contrastive estimation** Noise-contrastive estimation (NCE, Gutmann and Hyvärinen, 2010) spans a broad class of learning algorithms, at the core of which is negative sampling (Mikolov et al., 2013), i.e., learning an implicit metric space from positive and negative examples. Given reference samples, samples from a positive distribution (i.e., high similarity with reference points) and samples from a negative distribution (i.e., low similarity with reference points), contrastive learning methods learn an embedding

---

[1] $\Delta(\mathcal{X})$ denotes the entire set of distributions over the space $\mathcal{X}$.

where positive examples are located closer to the reference points than negative examples. One of the most well-known and commonly used NCE objectives is InfoNCE (Oord et al., 2018), which solves

$$\max_{\phi,\psi\in\Phi}\mathbb{E}_{x,y,y}\left[\log\frac{e^{\phi(x)^\top\psi(y)}}{\sum_{y'\in y\cup y}e^{\phi(x)^\top\psi(y')}}\right] \qquad (5)$$

over some hypothesis class $\Phi:\{\phi:\mathcal{X}\to\mathcal{Z}\}$ for input space $\mathcal{X}$, latent space $\mathcal{Z}$, $x\sim\mathbb{P}(\mathcal{X})$, $y\sim\mathbb{P}_{\text{positives}}(\mathcal{X})$ and $y\sim\mathbb{P}_{\text{negatives}}(\mathcal{X})$. Contrastive learning has been widely studied in the static unsupervised/ supervised learning settings (Hjelm et al., 2018; Chen et al., 2020; He et al., 2020), as well as in reinforcement learning (Kim et al., 2018; Mazoure et al., 2020) for learning state representations with desirable properties such as alignment and uniformity (Wang and Isola, 2020).

Solving Equation (5) for $(\phi^*,\psi^*)$ yields a critic $f : \mathcal{X} \times \mathcal{Y} \to \mathbb{R}$ which decomposes as $f^*(x,y)=\phi^*(x)^\top\psi^*(y)$ and, at optimality[2], captures the log-ratio of $\mathbb{P}_{\text{positives}}(\mathcal{X})$ and $\mathbb{P}_{\text{negatives}}(\mathcal{X})$:

$$f^*(x,y)\propto\log\frac{\mathbb{P}[y|x]}{\mathbb{P}[y]} . \qquad (6)$$

**Implicit dynamics models via NCE.** Various prior works (Du and Mordatch, 2019; Mazoure et al., 2020; Nachum and Yang, 2021) have studied the use of NCE to approximate a single-step dynamics model, where triplets $(s_t,a_t,s_{t+1})$ have higher similarity than $(s_t,a_t,s_{t'\neq t+1})$, effectively defining positive and negative distributions over trajectory data. More recently, contrastive goal-conditioned RL (Eysenbach et al., 2022) used InfoNCE to condition the critic on goal states sampled from the replay buffer. These methods use asymetric encoders, using $\phi(s_t,a_t)$ and $\psi(s_{t+\Delta t})$, where positive samples of $s_{t+\Delta t}$ are sampled from the discounted state occupancy measure for $t\geq 0$.

The conditional probability distribution of future states given the current state-action pair can be efficiently estimated using an implicit model trained via contrastive learning over positive and negative feature distributions, as shown in Equation (7).

$$\ell_{\text{InfoNCE}}(\phi,\psi)=\mathbb{E}_{s_t,a_t,\Delta t,\Delta t}\left[-\log\frac{e^{\phi(s_t,a_t)^\top\psi(s_{t+\Delta t})}}{\sum_{\Delta t'\in\Delta t\cup\Delta t}e^{\phi(s_t,a_t)^\top\psi(s_{t+\Delta t'})}}\right] \qquad (7)$$

Minimizing $\ell_{\text{InfoNCE}}$ over trajectory data yields a critic which, at optimality, approximates the future discounted state occupancy measure up to a multiplicative term as per Equation (6),

$$f^*(s_t,a_t,s_{t+\Delta t})\propto\log\frac{\mathbb{P}[s_{t+\Delta t}|s_t,a_t;\pi]}{\mathbb{P}[s_{t+\Delta t};\pi]} . \qquad (8)$$

Intuitively, $f^*$ approximates a $H$-step dynamics model which has an implicit dependence on policy $\pi$ that collected the training data, but is time-independent since Equation (8) is optimized on average across multiple $t,\Delta t$. Ordinarily, training state-space models is hard when the dimensions are large, e.g. image-based domains. However, by using contrastive learning, we can learn this model without having to require it predict high-dimensional observations, as similarity is evaluated in a lower-dimensional latent space (observe that in Equation (7) the inner product is computed in $\mathcal{Z}$, whose dimension we control, instead of $\mathcal{X}$, which is specified externally). An apparent limitation of the approach is that the probability of future states $s_{t+\Delta t}$ is recovered only up to a constant. However, it turns out that we can still use this model to get accurate estimates of the Q-values, as is described in the next section.

## 4 ESTIMATING AND MAXIMIZING RETURNS VIA CONTRASTIVE LEARNING

In this section, we show how NCE can be used to learn a quantity proportional to a value function, and how the later can be used in a policy iteration scheme.

### 4.1 ESTIMATING Q-VALUES USING THE CONTRASTIVE MODEL

As shown in Equation (3), the Q-function at $(s_t,a_t)$ can be thought of as evaluating the reward function at states sampled from the discounted occupancy measure $\mathbb{P}^\pi_{t:H}(s_t,a_t)$. That is, to estimate a quantity

---

[2]See Ma and Collins (2018) for exact derivation.

akin to $Q^\pi$, we can first estimate the occupancy measure and take a weighted average of rewards over future states using the probabilities from the log-density ratio learned by the contrastive model. Precisely, Equation (3) corresponds to using an importance-weighted estimator, where an optimal critic that minimizes Equation (7) approximates the density ratio from Equation (8). The positive samples come from the discounted state occupancy measure: we first sample a time offset $\Delta t \sim \text{Geometric}(1-\gamma)$ (column in the dataset), and then sample a state from the distribution of states at this given offset (row in the dataset). As per classical InfoNCE formulation, this forms the joint distribution $(s_t, a_t, s_{t+\Delta t})$, which is contrasted against the negative distribution of product of marginals $p(s_t, a_t) \times p(s_{t+\Delta t})$.

The critic itself can be trained using the occupancy measure formulation specified in Equation (4) over all state-action pairs in a given episode. However, Equation (4) needs to be re-adjusted to account for finite-horizon truncation of the geometric mass function presented in Definition 1.

**Definition 1 (Truncated distribution)** *Let $X$ be a random variable with distribution function $F_X$. $Y$ is a called the **truncated distribution** of $X$ with support $[m, M]$ s.t. $0 < m < M$ if*

$$\mathbb{P}[Y = y] = \frac{F_X(y-m) - F_X(y-1-m)}{F_X(M) - F_X(m)}, y = m, m+1, m+2, .. M \tag{9}$$

*We denote the special case of the truncated geometric distribution as $\text{TruncGeom}(p, m, M)$.*

The contrastive objective to train the critic to approximate the discounted occupancy measure over a dataset $\mathcal{D}$ is then

$$\ell_{\text{OM-InfoNCE}}(\phi, \psi) = \mathbb{E}_{\substack{s_t, a_t \sim \mathcal{D}, \\ \Delta t \sim \text{TruncGeom}(1-\gamma, t, H), \\ \Delta t \sim \text{TruncGeom}(1-\gamma, t' \neq t, H)}} \left[ -\log \frac{e^{\phi(s_t, a_t)^\top \psi(s_{t+\Delta t})}}{\sum_{\Delta t' \in \Delta t \cup \Delta t} e^{\phi(s_t, a_t)^\top \psi(s_{t+\Delta t'})}} \right] \tag{10}$$

It is possible that multiple optimal critics exist s.t. the multiplicative proportionality constant depends on the action. To avoid this, we adopt a similar approach as Eysenbach et al. (2022) and introduce a regularization term over the partition function, making the critic training objective be

$$\ell_{\text{Critic}} = \ell_{\text{OM-InfoNCE}} + \lambda_{\text{Partition}} \mathbb{E}_{s_t, a_t, \Delta t, \Delta t} [(\log \sum_{\Delta t' \in \Delta t \cup \Delta t} e^{\phi(s_t, a_t)^\top \psi(s_{t+\Delta t'})})^2] \tag{11}$$

Now, suppose we found an optimal critic $f$. Combining Equation (4) with Definition 1, we obtain the following form of the Q-function for an optimal critic $f$ which minimizes Equation (7):

$$Q_{\text{NCE}}(s_t, a_t) = \sum_{\Delta t=1}^{\infty} \gamma^{\Delta t-1} \int_{s_{t+\Delta t}} r(s_{t+\Delta t}) \mathbb{P}[s_{t+\Delta t} | s_t, a_t; \pi] ds_{t+\Delta t}$$

$$\propto \frac{1}{1-\gamma} \mathbb{E}_{\Delta t \sim \text{TruncGeom}(1-\gamma, t, H)} \left[ \int_{s_{t+\Delta t}} r(s_{t+\Delta t}) e^{f(s_t, a_t, s_{t+\Delta t})} \mathbb{P}[s_{t+\Delta t}; \pi] ds_{t+\Delta t} \right] \tag{12}$$

$$= \frac{1}{1-\gamma} \mathbb{E}_{\Delta t \sim \text{TruncGeom}(1-\gamma, t, H)} [\mathbb{E}_{\mathbb{P}_{t+\Delta t}^\pi} [r(s_{t+\Delta t}) e^{f(s_t, a_t, s_{t+\Delta t})}]]$$

Here, the offset $\Delta t$ is a random variable sampled from $\text{TruncGeom}(1-\gamma, t, H)$ where $H$ is the horizon of the MDP[3]. We can also show that $Q(s, a) < Q(s, a') \implies Q_{\text{NCE}}(s, a) < Q_{\text{NCE}}(s, a')$ for all $s \in \mathcal{S}$ and $a, a' \in \mathcal{A}$, which makes the contrastive Q-values suitable for policy evaluation. However, we do not, in general, expect $Q_{\text{NCE}}$ to recover the optimal Q function, as the recovered Q-values are on-policy with respect to $\pi$.

## 4.2 EFFICIENT ESTIMATION USING RANDOM FOURIER FEATURES

A major issue with using $Q_{\text{NCE}}$ out-of-the-box is that it is computationally expensive, requiring evaluation of the inner product $\phi(s_t, a_t)^\top \psi(s_{t+\Delta t})$ with a large number of future states and hence multiple forward passes through $\psi$. The underlying cause of this computational overhead is the RBF kernel term $e^{\phi(s_t, a_t)^\top \psi(s_{t+\Delta t})}$. If we instead used a linear kernel, the constant term $\phi(s_t, a_t)$ would be factored out, and we could separately keep track of reward-weighted future expected features. This would (1) reduce

---

[3]While using the truncated geometric distribution makes Equation (12) proportional to the true value function, the relation becomes an equality in the infinite-horizon case since $\lim_{H \to \infty} 1 - (1-p)^{H-m} = 1$.

the computational complexity of $N$ actor updates over $\mathcal{D}$ from $\mathcal{O}(|\mathcal{D}|N)$ to $\mathcal{O}(|\mathcal{D}|+N)$ and (2) reduce the variance of the representation if averaging features of future states using exponential moving average. It turns out that the RBF kernel can be approximately linearized by using random Fourier features (Rahimi and Recht, 2007; Nachum and Yang, 2021).

**Lemma 1** *Let $x, y \in \mathbb{R}^d$ be unit vectors, and let $F_{\mathbf{W}, b}(x) = \sqrt{\frac{2e}{d}} \cos(\mathbf{W}x + b)$ where $\mathbf{W} \sim Normal(0, \mathbf{I})$ and $b \sim Uniform(0, 2\pi)$* *fixed at initialization**. Then, $\mathbb{E}[F_{\mathbf{W}, b}(x)^\top F_{\mathbf{W}, b}(y)] = e^{x^\top y}$.*

Lemma 1 is a straightforward modification of the result from Rahimi and Recht (2007) and allows us to reduce the RBF kernel to an expectation over $d$-dimensional random feature vectors:

$$
\begin{aligned}
Q_{\text{NCE}}(s_t, a_t) &= \frac{1}{1-\gamma} \mathbb{E}_{\Delta t \sim \text{TruncGeom}(1-\gamma, t, H)} [\mathbb{E}_{\mathbb{P}(s_{t+\Delta t}; \pi)} [e^{\phi(s_t, a_t)^\top \psi(s_{t+\Delta t})} r(s_{t+\Delta t})]] \\
&= \frac{1}{1-\gamma} F_{\mathbf{W}, b}(\phi(s_t, a_t))^\top \mathbb{E}_{\Delta t \sim \text{TruncGeom}(1-\gamma, t, H)} [\mathbb{E}_{\mathbb{P}(s_{t+\Delta t}; \pi)} [F_{\mathbf{W}, b}(\psi(s_{t+\Delta t})) r(s_{t+\Delta t})]] \\
&= \frac{1}{1-\gamma} F_{\mathbf{W}, b}(\phi(s_t, a_t)) \xi(\pi)
\end{aligned}
\tag{13}
$$

The advantage of using the RFF approximation is that it allows us to split the exponential term inside the expectation and separately keep track of the policy-dependent, reward-weighted future state probability term, while the state-action dependence term is learned online. Specifically, we keep track of $\xi(\pi)$ via an exponential-moving average during the entire duration of training[4].

## 4.3 LEARNING THE POLICY

Once the policy evaluation phase completes and we have an estimate $Q_{\text{NCE}}$, we optimize a policy to maximize the returns predicted by this Q-value. We can decode the policy by minimizing its Kullback-Leibler divergence to the Boltzmann Q-value distribution (see Haarnoja et al. (2018)), which can be efficiently done by minimizing the following objective:

$$
\ell_{\text{Policy}}(\theta) = \mathbb{E}_{s_t \sim \mathcal{D}} \left[ D_{\text{KL}} \left( \pi_\theta(s_t) \middle\| \frac{e^{Q(s_t, \cdot)/\tau}}{\int_{a \in \mathcal{A}} e^{Q(s_t, a)/\tau} da} \right) \right].
\tag{14}
$$

Note that in discrete action spaces, minimizing Equation (14) leads to a soft version of the greedy policy decoding $\pi_{\text{greedy}}(s) = \arg\max_{a \in \mathcal{A}} Q_{\text{NCE}}(s, a)$ for $s \in \mathcal{S}$. In practice, we approximate the KL term in Equation (14) using $N_a$ Monte-Carlo action samples.

Decoding $\pi$ in such a way can lead to sampling out-of-distribution actions in regions where the Q-function might be inaccurate due to poor dataset coverage. To mitigate this issue, we follow prior work (Cobbe et al., 2021; Zhao et al., 2021; Schwarzer et al., 2021) and add a behavior cloning term which prevents the new policy from straying too far away from the data:

$$
\ell_{\text{BC}}(\theta) = \mathbb{E}_{a, s \sim \mathcal{D}} [\log \pi_\theta(a|s)] + \tau \mathbb{E}_{s \sim \mathcal{D}} [\mathcal{H}(\pi_\theta(s))].
\tag{15}
$$

for entropy estimator $\mathcal{H}(\pi(s)) = -\mathbb{E}_{a \sim \pi(s)}[\log \pi(a|s)]$. We add this extra loss to $\ell_{\text{Policy}}$ to learn a policy $\pi$ which prioritizes high Q-values that are well-supported by the offline dataset $\mathcal{D}$. Thus, the final policy optimization objective becomes

$$
\ell_{\text{Policy}}(\theta) = \ell_{\text{Policy}}(\theta) + \lambda_{\text{BC}} \ell_{\text{BC}}(\theta).
\tag{16}
$$

The policy found by minimizing $\ell_{\text{Policy}}$ has, on average, non-decreasing returns, as per Lemma 2.

**Lemma 2 (Contrastive policy improvement)** *Let $\mu$ be a policy and let $Q_{NCE}^\mu = \min_{\phi, \psi \in \Phi} \mathbb{E}_{\mathcal{D}^\mu}[\ell_{Critic}(\phi, \psi)]$. If*

$$
\pi(s) = \underset{\pi \in \Pi}{\arg\min} D_{KL} \left( \pi(s) \middle\| \frac{e^{Q_{NCE}^\mu(s_t, \cdot)/\tau}}{\int_{a \in \mathcal{A}} e^{Q_{NCE}^\mu(s_t, a)/\tau} da} \right)
\tag{17}
$$

*then $Q^\pi(s, a) \geq Q^\mu(s, a)$ for all $(s, a) \in \mathcal{D}^\mu$.*

---

[4]This idea can be adapted to online learning settings as well by clipping policy improvement steps so that $\xi$ doesn't change too fast under newly collected data.

---

**Algorithm 1:** Contrastive Value Learning (CVL)

---

**Input** : Dataset $\mathcal{D} \sim \mu$, $\psi$,$\phi$ networks, temperature parameter $\tau$, exponential moving average
parameter $\beta$

1 **for** *epoch* $j = 1,2,...,J$ **do**

2     **for** *minibatch* $\mathcal{B} \sim \mathcal{D}$ **do**

        /* Update density ratio estimator using Equation (11)     */

3         Update $\phi^{(j+1)}$,$\psi^{(j+1)}$ using $\nabla_{\phi,\psi}\ell_{\text{Critic}}(\phi^{(j)},\psi^{(j)})$ ;

        /* Estimate the contrastive Q-function       */

4         $Q(s,a) \leftarrow$ Equation (13) if using RFF, otherwise Equation (12);

        /* Decode policy from Q-function using Equation (16)     */

5         Update $\pi_\theta$ using $\nabla_\theta\{\ell_{\text{Policy}}(\theta)\}$ ;

        /* Update future state encoder using EMA       */

6         $\psi_{\text{EMA}}^{(j+1)} \leftarrow \beta\psi^{(j+1)} + (1-\beta)\psi_{\text{EMA}}^{(j)}$ ;

        /* Update future state features weighted by rewards     */

7         $\xi_{\text{EMA}}^{(j+1)} \leftarrow \psi_{\text{EMA}}^{(j+1)} \cdot \mathcal{B}[r_{t+\Delta t}]$;

---

The proof of Lemma 2 is located in Section 6.2. Specifically, Lemma 2 tells us that using CVL as a surrogate Q-function corresponds to one step of conservative policy improvement, where $\pi$ satisfies soft constraints of Equation (14) and small $\mathbb{E}_{\mathcal{D}^\mu}[D_{\text{KL}}(\pi(s)||\mu(s))]$ via the BC term.

### 4.4 PRACTICAL IMPLEMENTATION

We now present our complete method, which can be viewed as an actor-critic method for offline RL. We learn the critic via contrastive learning (Equation (11)) and learn the policy via Equation (16). We will interleave these steps in most of our experiments, but experiments in Section 5.1 show that the critic can be pretrained e.g. in the presence of unlabeled data from related tasks. We summarize the method in Algorithm 1.

### 4.5 INTERPRETATIONS AND CONNECTIONS WITH PRIOR WORK

The main distinction between Contrastive Value Learning and prior works consists specifically in representing the Q-values in a two-step decomposition: the Q-value is represented as an occupancy measure weighted by the reward signal; the occupancy measure itself is represented using a powerful likelihood-based model parameterized using an implicit function. Decoupling the learning of the occupancy measure from reward maximization allows, among others, for efficient pretraining strategies on unlabeled data, i.e. trajectory data without reward information, and can be used to learn provably optimal state representations for *any* reward function (Touati and Ollivier, 2021). While CVL is similar in spirit to the successor representation (Dayan, 1993; Barreto et al., 2016), the occupancy measure learned by CVL is much richer than that of SR, as it captures the entire distribution over future states instead of only the first moment. Another method, $\gamma$-models (Janner et al., 2020), is closely related to CVL, but uses a surrogate single-step TD objective to learn the occupancy measure, similarly to C-learning (Eysenbach et al., 2020).

## 5 EXPERIMENTS

Our experiments aim to answer three questions. First, we study how CVL compares with baseline approaches on a large benchmark of state-based tasks. Our second set of experiments look at image-based tasks, testing the hypothesis that CVL scales to these tasks more effectively than the baselines. We conclude with ablation experiments. Our main point of comparison will be a high-performing offline RL method, CQL (Kumar et al., 2020). While CVL learns an implicit model, that model is structurally much more similar to value-based RL methods than model-based methods, motivating our comparison to a value-based baseline (CQL). We will also include behavioral cloning as a baseline.

**Metaworld.** We first test our approach on the complex MetaWorld benchmark (Yu et al., 2020a), which consists of 50 robotic manipulation tasks such as open a door, pick up an object, reach a certain area of the table, executed by a robotic arm (see Figure 2 (left)). This domain is an ideal testbed for CVL, as it allows

Figure 2: **Metaworld benchmark.** *(Left)* We evaluate CVL on 50 tasks from Metaworld, a subset of which are shown here. *(Right)* Compared with three offline RL baselines, CVL achieves statistically-significant improvements in offline performance. Results are reported over 5 random seeds.

for both full-state and image-based experiments, has a dense and informative reward function thus decoupling the problem of representation learning from exploration, and is challenging for model-free methods which leaves room for improvement. While the original MetaWorld domain has been used to evaluate online RL agents, we create an *ad hoc* dataset suitable for offline learning. To do so, we train Soft Actor-Critic (Haarnoja et al., 2018) from full states on each of the 50 tasks separately for 500k frames, and save the resulting replay buffer, which forms the training dataset. As shown in Figure 2 (right), CVL manages to considerably improve upon strong baselines such as behavior cloning, CQL and CQL with UDS (Yu et al., 2022)[5]. We report the results on all tasks of the MetaWorld suite over 5 random seeds, according to the aggregation methodology proposed by Agarwal et al. (2021b). Per-environment scores are available in Table 4.

**Image-based experiments** Our working hypothesis is that contrastive formulation of the value function acts in itself as a pre-training mechanism via the prism of representation learning. For this reason, we conduct further experiments on 4 image-based tasks from the MetaWorld suite (similarly to full-states, the dataset was obtained from the SAC replay buffer trained on rendered images).

Table 1: **Offline RL with Images.** We compare CVL to baselines on four offline, image-based tasks from MetaWorld offline image-based tasks. Average $\pm$ std. dev. are shown for 5 random seeds.

| Task | BC | CQL | CVL |
|---|---|---|---|
| door-close | $571 \pm 9.9$ | $4249 \pm 269.9$ | $\mathbf{4480 \pm 305.1}$ |
| door-open | $178 \pm 4.0$ | $2099 \pm 0.9$ | $\mathbf{3389 \pm 76.6}$ |
| drawer-close | $2414 \pm 1736.5$ | $\mathbf{3964 \pm 1634.9}$ | $2177 \pm 1679.5$ |
| drawer-open | $1030 \pm 104.2$ | $820 \pm 56.0$ | $\mathbf{2543 \pm 115.0}$ |

Results presented in Table 1 show that CVL is also able to learn meaningful Q-values and achieve good empirical performance on hard image-based tasks.

## 5.1 ABLATION EXPERIMENTS.

**When is pretraining the model useful?** In theory, the model can be pretrained on the data from other tasks, however, we do not always expect this to help (e.g., when the pretraining tasks are very different). We ran an experiment to test this capability. The results, shown in Fig. 3, show that pretraining sometimes speeds up learning. In particular, we observe that pretraining is effective when the pretraining tasks are similar to the target task and contain a diverse set of state-action pairs.

**How reliable is the $Q_{\text{NCE}}$ approximation?** Given that contrastive Q-values are proportional to the true Q-function, a natural question to ask is how good is $Q_{\text{NCE}}$ at capturing the topology of $Q$? First, we conduct an ablation demonstrating how linearizing the RBF kernel via random Fourier features provides a performance gain on the offline MetaWorld tasks Figure 4. Specifically, we hypothesize that this is due to the reduced variance of the RFF Q-value estimator which keeps track of future reward-weighted state features using a rolling average.

Next, we qualitatively assess the similarity between contrastive and true Q-values on the continuous Mountain Car environment (Moore, 1990) by first pre-training SAC online on the task and then fitting CVL to the data from SAC's replay buffer. Figure 5 (left) shows the contrastive Q-values on a log-scale, evaluated on trajectories from the SAC replay; for comparison, we also show the Q-values learned by online SAC in Figure 5 (right). Note that the value function learned by CVL conserves the same topology as the true value function, up to a multiplicative rescaling.

---

[5]For CQL+UDS, we combine all data from the current task with unlabeled data from related tasks with rewards set to 0. In the absence of related tasks, we pre-train the critic on the current task with 0 rewards.

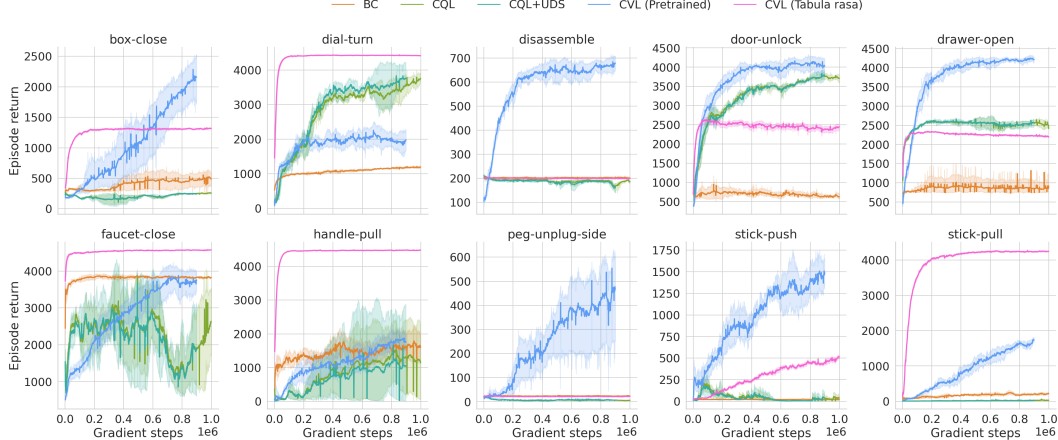

Figure 3: **Offline Learning Curves for Metaworld.** Episode return curves as a function of gradient steps taken during training on 10 random MetaWorld tasks; curves show mean $\pm$ standard deviation. Pretraining the reward-free occupancy measure on related tasks allows CVL to outperform baseline approaches and even CVL trained *tabula rasa*.

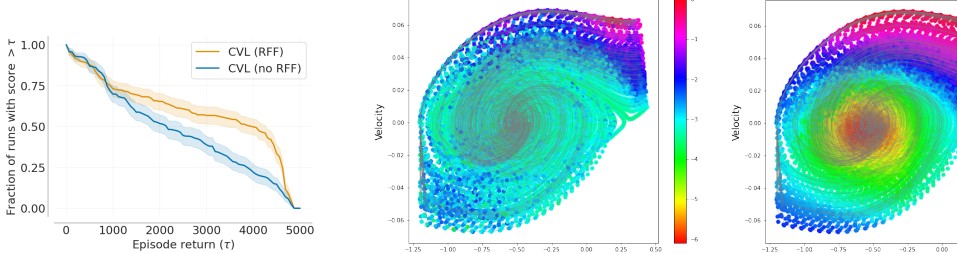

Figure 4: CVL with RFF (orange) performs slightly better than without RFF (blue).

Figure 5: **Visualizing the estimated Q-values.** *(Left)* Normalized $\log Q_{\mathrm{NCE}}$ learned by CVL offline on the Mountain Car environment. *(Right)* Normalized $Q$ learned by online SAC on the same environment.

## 6   DISCUSSION

This paper presented an RL algorithm that learns a contrastive model of the world, and uses that model to obtain Q values by estimating the likelihood of visiting future states. Our experiments demonstrate that this approach can effectively solve a large number of offline RL tasks, including from image-based observations. Our pretraining results hinted that CVL can be pretrained on datasets from other tasks, and we are excited to pretrain our model on datasets of increasing size.

**Limitations.**   One limitation of our approach is that it corresponds to a single step of policy improvement. This limitation might be lifted by training the contrastive model using a temporal difference update for the contrastive model (Eysenbach et al., 2020; Blier et al., 2021). A second limitation is that the RFF approximation can be poor when the feature dimension is small. We tried to train the contrastive model using non-exponentiated features (akin to HaoChen et al. (2021)), but failed to achieve satisfactory results. Figuring out how to effectively train these spectral models remains an important question.

## REPRODUCIBILITY STATEMENT

We ensure reproducibility of our method via a) releasing the offline MetaWorld dataset that we used for our main results to allow the community to conduct further research on this domain upon publication, b) releasing the code used to obtain our results upon publication and c) detailing the hyperparameters and design choices for the implementation of CVL and the computational resources used for our experiments in Section 6.1.

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

APPENDIX

REPRODUCIBILITY CHECKLIST


## 6.1 EXPERIMENTAL DETAILS

**Model architecture**   All algorithms (baselines as well as CVL) were based on a common architecture, where an encoder (IMPALA (Espeholt et al., 2018) for image data and two layer DenseNet MLP (Huang et al., 2017) for full-states) generated state features which, combined with actions gave rise to the Q-value and the policy (we used a diagonal Gaussian policy with a Tanh bijector, as is common for continuous control tasks). The main difference of CVL with the baselines is that the critic is defined implicitly via the dot-product of current state-action features passed through one encoder, and future state features passed into a separate DenseNet. The output of both encoders was optionally normalized using $\ell_2$ norm. All methods had a LayerNorm layer (Ba et al., 2016) in between each linear layer to ensure proper feature scaling.

| Hyperparameter | Value |
|---|---|
| Learning rate | $3 \times 10^{-4}$ |
| Batch size (all but CVL) | 512 |
| Discount factor | 0.99 |
| Framestack | No |
| Max gradient norm | 100 |
| MLP structure | $256 \times 256$ DenseNet |
| Encoder (full-state) | $256 \times 256$ DenseNet MLP |
| Encoder (pixels) | IMPALA |
| Add LayerNorm in between all layers | Yes |

Table 2: Hyperparameters that are consistent between methods.

| Hyperparameter | Value |
|---|---|
| **CVL** | |
| Batch size | $H$ |
| Number of future action samples $N_a$ | 10 |
| InfoNCE temperature | 1 |
| Partition function coefficient $\lambda_{\text{Partition}}$ | 0.001 |
| BC coefficient $\lambda_{\text{BC}}$ | 0 (Mountain Car), 0.1 (rest) |
| RFF | Yes |
| $\ell_2$-normalize MLP outputs | Yes |
| **CQL** | |
| Regularization coefficient | 1 |
| **BC** | |
| Entropy regularization coefficient | 0.1 |

Table 3: Hyperparameters that are different between methods.

All experiments were run on the equivalent of 8 V100 GPUs with 64 Gb of RAM and 8 CPUs. For all methods, the corresponding auxiliary loss weights have been selected through best aggregated performance on the `drawer` and `door` domains with hyperparameter values of $\{0, 0.01, 0.1, 1.0\}$.

**Dataset composition**   The offline MetaWorld dataset was constructed by first pre-training SAC on all 50 tasks from full-states for 500k environment interactions. The replay buffer at the end of the training was then used as training dataset for BC, CQL, CQL+UDS and CVL. An identical approach was used to construct the image-based MetaWorld datasets and the Mountain Car dataset.

**Pretraining setup**   When pretraining CVL, we first optimize the critic on unlabeled data from dataset for all the semantically related tasks, i.e. tasks which belong to the same domain, and then finetune both the critic and the policy on reward-labeled data from the target task. Semantically related tasks in MetaWorld are easily identifiable by their domain name, e.g. `drawer-open` and `drawer-close` belong to

the `drawer` domain. We use a similar approach when pretraining CQL+UDS, where we perform TD updates with all rewards equal to 0 during the pretraining phase.

## 6.2 PROOFS

**Proof 1 (Random Fourier features approximation, Lemma 1)**

For unit vectors $x, y \in \mathbb{R}^d$, $d > 0$,

$$
\begin{aligned}
\mathbb{E}\Big[\Big(\sqrt{\tfrac{2}{d}}\cos(Wx+b)\Big)^\top \Big(\sqrt{\tfrac{2}{d}}\cos(Wy+b)\Big)\Big] &= \exp\{-\|x-y\|_2^2/2\} \\
&= \exp\{-(\|x\|_2^2 - 2x^\top y + \|y\|_2^2)/2\} \\
&= \exp\big(-(2-2x^\top y)/2\big) \\
&= e^{x^\top y - 1} \\
&= \frac{e^{x^\top y}}{e}
\end{aligned}
\tag{18}
$$

by re-arranging the terms in the result from Rahimi and Recht (2007). Therefore,

$$
e^{x^\top y} = \mathbb{E}\Big[\Big(\sqrt{\tfrac{2e}{d}}\cos(Wx+b)\Big)^\top \Big(\sqrt{\tfrac{2e}{d}}\cos(Wy+b)\Big)\Big]
\tag{19}
$$

**Proof 2 (CVL induces a single-step of policy improvement, Lemma 2)** *Since, for the optimal critic $f^*$,*

$$
e^{f^*(s_t, a_t, s_{t+\Delta t})} \propto \frac{\mathbb{P}[s_{t+\Delta t}|s_t, a_t; \mu]}{\mathbb{P}[s_{t+\Delta t}; \mu]}.
\tag{20}
$$

*point-wise for every $(s_t, a_t, s_{t+\Delta t}) \in \mathcal{D}^\mu$, then, for $\alpha > 0$,*

$$
e^{f^*(s_t, a_t, s_{t+\Delta t})} = \alpha \frac{\mathbb{P}[s_{t+\Delta t}|s_t, a_t; \mu]}{\mathbb{P}[s_{t+\Delta t}; \mu]}.
\tag{21}
$$

*Now, the following relation holds using the previous result*

$$
\begin{aligned}
Q_{NCE}^\mu(s_t, a_t) &= \frac{1}{1-\gamma}\mathbb{E}_{\Delta t \sim TruncGeom(1-\gamma, t, H)}[\mathbb{E}_{\mathbb{P}_{t+\Delta t}^\mu}[r(s_{t+\Delta t})e^{f(s_t, a_t, s_{t+\Delta t})}]] \\
&= \frac{\alpha}{1-\gamma}\mathbb{E}_{\Delta t \sim TruncGeom(1-\gamma, t, H)}\Big[\int_{s_{t+\Delta t}} r(s_{t+\Delta t})\mathbb{P}[s_{t+\Delta t}|s_t, a_t; \mu]ds_{t+\Delta t}\Big] \\
&= \alpha Q^\mu(s_t, a_t)
\end{aligned}
\tag{22}
$$

*Using this relation yields*

$$
\frac{e^{Q_{NCE}^\mu(s_t, a_t)/\tau}}{\int_{a \in \mathcal{A}} e^{Q_{NCE}^\mu(s_t, a)/\tau}da} = \frac{e^{\alpha Q^\mu(s_t, a_t)/\tau}}{\int_{a \in \mathcal{A}} e^{\alpha Q^\mu(s_t, a)/\tau}da} = \frac{e^{Q^\mu(s_t, a_t)/\tau}}{\int_{a \in \mathcal{A}} e^{Q^\mu(s_t, a)/\tau}da}
\tag{23}
$$

*It follows that*

$$
\begin{aligned}
\operatorname*{argmin}_{\pi \in \Pi} D_{KL}\Big(\pi(s_t)\Big\|\frac{e^{Q_{NCE}^\mu(s_t, \cdot)/\tau}}{\int_{a \in \mathcal{A}} e^{Q_{NCE}^\mu(s_t, a)/\tau}da}\Big) &= \operatorname*{argmin}_{\pi \in \Pi} D_{KL}\Big(\pi(s_t)\Big\|\frac{e^{Q^\mu(s_t, \cdot)/\tau}}{\int_{a \in \mathcal{A}} e^{Q^\mu(s_t, a)/\tau}da}\Big) \\
&= \pi(s_t)
\end{aligned}
\tag{24}
$$

*Now, we invoke Lemma 2 from Haarnoja et al. (2018) by using the equivalence of the policy decoded from contrastive Q-values to the policy found by soft policy iteration, which concludes the proof.*

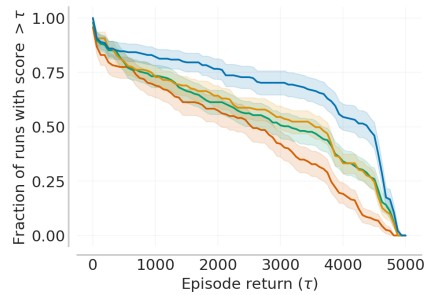

Figure 6: Performance profile of BC (red), CQL (green), CQL+UDS (orange) and CVL (blue) generated by the rliable library (Agarwal et al., 2021b) for the offline MetaWorld experiments over 5 random seeds.

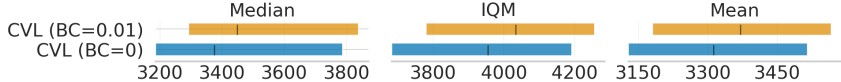

Figure 7: Aggregated performance metrics for CVL with different behavior cloning weights.

## 6.3 ADDITIONAL RESULTS

### 6.3.1 METAWORLD

**Ablation on the BC coefficient:** We ablate the impact of the behavior cloning loss on CVL's performance in Figure 7. We can see that, although adding a behavior cloning loss improves the performance by a small amount, it is not essential to the fundamental functioning of CVL.

### 6.3.2 MOUNTAIN CAR

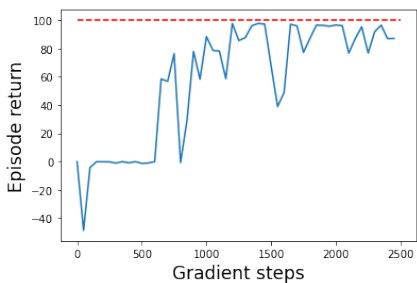

Figure 8: Evaluation returns on Mountain car during training on data from the SAC replay buffer. The red dotted line indicates highest possible return.

**Quantitative evaluation of the contrastive occupancy measure:** From Wang and Tay (2022), we know that

$$\mathrm{MMD}(\mathbb{P},\mathbb{Q}) \leq 2\sqrt{1-e^{-\mathrm{KL}(\mathbb{P},\mathbb{Q})}} \tag{25}$$

We also know that

$$\begin{aligned}\mathcal{I}(\mathbb{P},\mathbb{Q}) &= \mathrm{KL}((\mathbb{P},\mathbb{Q})\|\mathbb{P}\otimes\mathbb{Q}) \\ &\geq \log N - \ell_{\mathrm{InfoNCE}}(\mathbb{P}_N,\mathbb{Q}_N)\end{aligned} \tag{26}$$

which simplifies the above expression to

$$\hat{\mathrm{MMD}}_N(\mathbb{P},\mathbb{Q}) \leq 2\sqrt{1-e^{-\log N+\ell_{\mathrm{InfoNCE}}(\mathbb{P}_N,\mathbb{Q}_N)}} \tag{27}$$

Figure 9 shows the upper-bound on the MMD between occupancy measures learned with temporal difference and contrastive learning methods.

| Task | BC | CQL | CQL+UDS | CVL (Tabula rasa) | CVL (Pretrained) |
|---|---|---|---|---|---|
| basketball | 3188 ± 348.9 | 646 ± 2.2 | 678 ± 112.6 | **4503 ± 113.8** | 4171 ± 285.9 |
| bin-picking | 13 ± 1.5 | 28 ± 0.7 | 21 ± 1.9 | 18 ± 1.8 | **860 ± 69.2** |
| box-close | 891 ± 164.4 | 311 ± 12.4 | 296 ± 19.8 | 1496 ± 131.2 | **4189 ± 352.4** |
| button-press | 2667 ± 85.3 | 3445 ± 60.2 | 3420 ± 273.4 | **3659 ± 51.9** | 1906 ± 360.2 |
| button-press-topdown | 3089 ± 256.7 | 3406 ± 525.7 | 3505 ± 990.9 | **3889 ± 36.2** | 548 ± 49.5 |
| button-press-topdown-wall | 1692 ± 47.0 | 2095 ± 28.4 | **2135 ± 66.4** | 2008 ± 21.7 | 546 ± 90.8 |
| coffee-button | 3490 ± 1435.5 | 3655 ± 740.6 | 3431 ± 689.8 | **4259 ± 169.9** | 149 ± 10.8 |
| coffee-pull | 647 ± 11.7 | 250 ± 10.3 | 330 ± 3.2 | **833 ± 27.2** | 167 ± 0.6 |
| dial-turn | 1331 ± 48.7 | 4257 ± 389.4 | 4449 ± 276.1 | 4526 ± 42.8 | **4611 ± 176.9** |
| disassemble | 215 ± 4.3 | 215 ± 9.6 | 217 ± 36.0 | 214 ± 18.6 | **926 ± 5.6** |
| door-close | 3634 ± 141.5 | **4555 ± 200.9** | 4547 ± 215.2 | 4544 ± 7.6 | 4313 ± 194.0 |
| door-lock | 3073 ± 303.7 | 3775 ± 59.1 | **3777 ± 144.2** | 3537 ± 271.1 | 557 ± 20.6 |
| door-open | 828 ± 24.7 | 4526 ± 71.7 | **4531 ± 179.0** | 3985 ± 279.6 | 613 ± 113.0 |
| door-unlock | 1322 ± 181.1 | 4122 ± 50.2 | 4002 ± 80.1 | 3139 ± 413.7 | **4618 ± 49.7** |
| drawer-close | 4619 ± 53.4 | 4855 ± 0.0 | **4857 ± 2.0** | 4853 ± 6.8 | 2933 ± 671.8 |
| drawer-open | 1727 ± 204.0 | 2768 ± 45.6 | 2776 ± 25.2 | 2512 ± 149.3 | **4664 ± 14.4** |
| faucet-close | 4160 ± 49.8 | **4752 ± 1585.0** | 4713 ± 1724.2 | 4683 ± 47.8 | 4739 ± 57.0 |
| faucet-open | 2052 ± 80.9 | **4731 ± 401.8** | 4729 ± 561.1 | 3660 ± 221.9 | 1637 ± 64.9 |
| hammer | 2158 ± 272.0 | 898 ± 70.3 | 1030 ± 126.6 | **4632 ± 73.6** | 4630 ± 86.5 |
| hand-insert | 44 ± 17.8 | 443 ± 2.0 | 428 ± 1.5 | 180 ± 5.3 | **4612 ± 539.8** |
| handle-press | 4734 ± 36.3 | 2816 ± 4.4 | 2755 ± 0.8 | **4861 ± 28.6** | 2417 ± 169.2 |
| handle-press-side | 3820 ± 1556.5 | 4783 ± 170.5 | 4786 ± 478.1 | **4816 ± 352.6** | 654 ± 27.7 |
| handle-pull | 3642 ± 968.8 | 2422 ± 524.1 | 2436 ± 1286.8 | 4594 ± 38.6 | **4636 ± 35.8** |
| handle-pull-side | 3418 ± 1002.2 | 1898 ± 582.6 | 1757 ± 343.2 | **4660 ± 41.0** | 2904 ± 92.4 |
| lever-pull | 3659 ± 180.8 | 2233 ± 399.5 | 2157 ± 258.0 | **4459 ± 107.8** | 4207 ± 98.9 |
| peg-insert-side | 11 ± 1.1 | 17 ± 4.1 | **19 ± 1.8** | 15 ± 0.4 | 12 ± 0.8 |
| peg-unplug-side | 56 ± 1.9 | 29 ± 2.6 | 29 ± 2.4 | 87 ± 1.6 | **4593 ± 24.6** |
| pick-out-of-hole | 10 ± 0.2 | 207 ± 0.4 | 191 ± 3.4 | **1245 ± 186.4** | 5 ± 0.9 |
| pick-place | 1771 ± 416.2 | 1263 ± 407.6 | 1306 ± 128.5 | 2942 ± 454.1 | **4403 ± 508.3** |
| pick-place-wall | 0 ± 0.0 | 1 ± 0.0 | 71 ± 0.0 | 19 ± 0.0 | **3522 ± 775.3** |
| plate-slide | 3979 ± 57.3 | 2697 ± 475.3 | 3508 ± 747.0 | **4649 ± 142.6** | 802 ± 12.4 |
| plate-slide-back | 2402 ± 333.9 | 3163 ± 1290.3 | 3014 ± 303.9 | **4718 ± 306.8** | 196 ± 5.0 |
| plate-slide-back-side | 4017 ± 874.6 | 4736 ± 1519.0 | 4732 ± 137.6 | **4752 ± 196.9** | 4669 ± 95.1 |
| plate-slide-side | 2241 ± 536.9 | **3104 ± 308.1** | 3015 ± 329.5 | 2695 ± 413.8 | 1939 ± 27.5 |
| push | 1834 ± 317.9 | 494 ± 5.0 | 463 ± 3.3 | 1997 ± 196.8 | **4386 ± 192.7** |
| push-back | 9 ± 0.2 | 71 ± 1.4 | 135 ± 0.8 | 109 ± 1.4 | **204 ± 20.9** |
| push-wall | 3327 ± 508.6 | 689 ± 5.6 | 628 ± 4.2 | 4502 ± 176.7 | **4601 ± 205.6** |
| reach | 3069 ± 359.2 | 3301 ± 920.3 | 3275 ± 677.8 | **4819 ± 182.9** | 4658 ± 204.8 |
| reach-wall | 4515 ± 93.9 | 4828 ± 26.5 | **4829 ± 49.1** | 4811 ± 27.0 | 4825 ± 21.2 |
| stick-pull | 595 ± 19.8 | 297 ± 2.0 | 441 ± 3.7 | **4488 ± 52.0** | 3434 ± 162.9 |
| stick-push | 263 ± 6.1 | 896 ± 3.9 | 897 ± 3.4 | 1155 ± 147.5 | **2804 ± 551.3** |
| sweep | 817 ± 124.3 | 3086 ± 645.7 | 3162 ± 1507.1 | 4127 ± 567.2 | **4461 ± 49.5** |
| sweep-into | 532 ± 151.3 | 1974 ± 34.0 | 1834 ± 870.1 | **2657 ± 364.3** | 506 ± 14.8 |
| window-close | 3739 ± 80.4 | 4478 ± 452.5 | 4442 ± 13.1 | **4534 ± 56.2** | 4519 ± 63.2 |
| window-open | 3743 ± 147.3 | 2773 ± 1433.5 | 2841 ± 1163.5 | **4534 ± 109.4** | 524 ± 320.0 |

Table 4: Evaluation returns on MetaWorld offline tasks. Average ± standard deviation are shown for 5 random seeds.

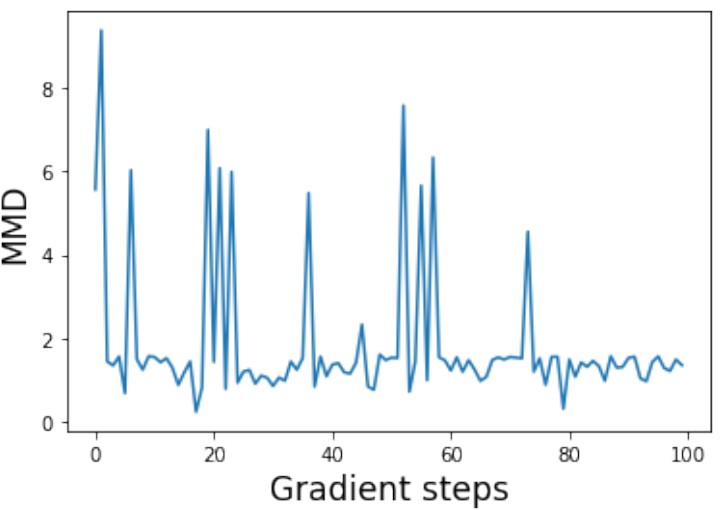

Figure 9: Upper-bound on the MMD between occupancy measures estimated via TD and contrastive learning.

