# OpenReview forum: "Contrastive Value Learning: Implicit Models for Simple Offline RL"
_ICLR.cc/2023/Conference — Submitted to ICLR 2023_

### Official Review · Reviewer_qU1j · 2022-10-24

**Confidence:** 3
**Correctness:** 2
**Technical Novelty And Significance:** 3
**Empirical Novelty And Significance:** 3
**Recommendation:** 3

**Clarity, Quality, Novelty And Reproducibility:**

By and large, I did not find the paper to be very clear. Especially, I found the introduction to not help much in understanding the actual contributions of the paper.

I believe there are some novel ideas, though, I am not all that familiar with the full literature on noise contrastive estimation in RL. As far as I know, expressing the Q-value in terms of a noise contrastive estimate of the state and action conditional future occupancy measure does not appear elsewhere.

I feel it would be very hard to reproduce the results from the description of the method in the paper, as many details are missing, however, the authors have stated that they will release the code upon publication.

Please see above for further details.

**Strength And Weaknesses:**

The idea is interesting, and the empirical results appear superficially promising. However, there appear to be a number of technical and clarity issues, and missing details which lead me to lean strongly towards rejecting the paper in its current state.

On the positive side, the derivation of the method for estimating action values using noise contrastive state representations in equation 12 is interesting and appears correct to me. I'm unsure of its practical utility given the contrastive objective only learns a ratio $p(s_{t+\Delta t}|s_t,a_t)/p(s_{t+\Delta t})$ (up to a multiplicative constant). Thus it seems like the method still requires sampling from $p(s_{t+\Delta t})$. Presumably, the variance would be high in many cases if the state space is large and the unconditioned occupancy measure is very different from the discounted occupancy following a particular state-action pair.

The description of the method in the introduction is a bit convoluted and, in my opinion, makes it harder rather than easier to understand what follows. In particular, phrasing like "we will learn an implicit model of the discounted state occupancy measure, which answers the question 'where will the agent be in the time-averaged future?'" gives the impression that the learned model is of a form that it can take a state action pair as input, and predicted the future state occupancy. Based on my understanding, the contrastive objective only learns a function $f^*(s_t,a_t,s_{t+\Delta t})$ such that $f^*(s_t,a_t,s_{t+\Delta t})\approx p(s_{t+\Delta t}|s_t,a_t)/p(s_{t+\Delta t})$ and thus can only really be used to correct samples from $p(s_{t+\Delta t})$ to samples from $p(s_{t+\Delta t}|s_t,a_t)$. While this is perhaps clarified in the phrase "treating it as a classifier rather than a generative model of observations", I feel like this could have been stated much more straightforwardly.

I am concerned with the correctness of equation 13 and the algorithm that is derived from it. On the left-hand side of the equation is a specific value, while the right-hand side is a random variable. This is in itself perhaps only a minor oversight if we assume the intended meaning is that the expectation of the RHS is equal to the LHS. However, $\xi(\pi)$, which is never formally defined, should also be a random variable as it depends on the randomly sampled W and b used in the RFF approximation. As far as I can tell, this is not accounted for as $\xi(\pi)$ is approximated using a running average, which presumably averages over many different values of W and b. I believe the RFF estimator in Lemma 1 is only correct if the same W and b are used in both factors so I'm not sure from the description how the algorithm actually works in practice or whether it is correct. Perhaps I overlooked something here, and the authors can explain in more detail how the algorithm works and why it is correct. At the very least I feel a more detailed explanation is required in the paper.

If I'm reading correctly, Figure 5 shows the log of the $Q_{NCE}$ values compared to the normalized SAC Q-values. Why is one on the log scale and the other isn't? I understood that the $Q_{NCE}$ values themselves should be proportional to the regular Q-values, not their log. Or is the intended meaning that both are on the log scale? Moreover, the claim that both figures show the same topology is not at all clear to me from the picture beyond very basic features around the edges.

Algorithm 1 is not entirely clear in a number of places. $\psi^{(j+1)}$ is updated both in line 3 by a gradient, and then in line 6 as an exponential moving average, is this really the intended meaning? I think the issue may be that $\psi^{(j+1)}$ is overloaded to mean both the exponential moving average and the immediate value. On line 7, what is the boldfaced $\beta$? I can't see that it has been defined anywhere. It is also stated that $\phi$ and $\psi$ are updated using the gradient of $l_{Critic}$ but presumably, this is done somehow using samples in the offline dataset as opposed to explicitly computing the gradient of the full expectation so more details here would be helpful. The same is true for the policy update on line 5.

While hyperparameters are provided in the appendix, it would help with the interpretation of the results to describe how they were chosen. i.e. tuned for the proposed approach, kept fixed from prior work. ect.

Other Minor Comments Questions and Suggestions
==============================================
* I don't understand footnote 3, I thought the reason for the proportionality is that $e^{f(s_t,a_t,s_{t+\Delta t})}$ is only equal to the suggested ratio up to a constant. Hence, I don't understand this claim.

* In section 4.2: "It is possible that multiple optimal critics exist s.t. the multiplicative proportionality constant depends on the action." Could the authors clarify this? If it can depend on the action, can it also depend on the state $s_t$? And if it can depend on both of these, what makes it a proportionality "constant" at all?

* I can't find it stated anywhere how long training proceeded in the main experiments. I see in Figure 3 that for the pretraining experiments a total of 1 million gradient steps are used, is this the same for all experiments and algorithms?

* Why is the batch size for CVL set to H according to table 3? Does this simply mean it trains on all the transitions in an episode at once? Furthermore, how does H compare to the batch size of 512 used by the other methods? Assuming training is done for a fixed number of gradient updates, this might be important in interpreting the results.

* The left-hand side of lemma 1 in the appendix should be in expectation.

* The first appearance of "occupancy measure" in the introduction is a bit jarring since it is never before mentioned that the paper will involve learning an occupancy measure

**Summary Of The Paper:**

This paper suggests an algorithm for offline reinforcement learning based on first applying a noise contrastive objective to estimate a quantity proportional to $p(s_{t+\Delta t}|s_t,a_t)/p(s_{t+\Delta t})$, where $\Delta t$ belongs to a geometric distribution of future times. That is, the ratio of the discounted future state occupancy conditioned on a state-action pair, to the unconditioned discounted future state occupancy, both under a particular policy. This estimate is then used to construct an estimator for action-values by reweighing a dataset of rewards and states generated under the current policy. Building on this idea, the paper also proposes an alternative action value estimator which decomposes the Q-value as a dot product of two vectors. The first vector depends on the policy and reward function and is tracked as a running average. The other vector depends on the state and action and is learned. This decomposition is motivated by the random features approach of Rahimi and Recht (2007). The proposed approaches are evaluated for offline RL mainly in simulated robotics tasks from MetaWorld and found to compare favourably to alternative approaches to offline RL. It is also demonstrated that when additional offline data without rewards is available, it can be use to pretrain $p(s_{t+\Delta t}|s_t,a_t)/p(s_{t+\Delta t})$, which in some cases provides a benefit.

**Summary Of The Review:**

The idea is superficially interesting, and the empirical results are superficially promising. However, there appear to be a number of technical and clarity issues, and missing details which lead me to lean strongly towards rejecting the paper in its current state. I am particularly concerned about the correctness of Equation 13 and the lack of detail and clarity in the pseudocode of algorithm 1. If the authors can address these issues, I would consider revising my score.

---

> ### Author Response · Authors · 2022-11-16
> **Response to reviewer qU1j**
>
> Dear reviewer,
>
> Thank you for your comments and for the suggestions about improving the paper. It seems like the reviewer’s main concern is with the correctness and clarity of the paper, which we have tried to address by revising every section noted by the reviewer.
>
> *Regarding RFF estimator of Lemma 1:* The reviewer is correct, since W and b have to be sampled once during the initialization phase. We have clarified this in the revised manuscript.
>
> *Regarding similarity of $Q^\mu$ and $Q^\text{NCE}$:* We have added Figure 9 which shows the upper-bound on the MMD between TD and contrastive occupancy measures as a function of training steps. As shown in Figure 5, both functions are proportional up to a multiplicative constant, which heavily affects the values in the middle of the $Q^\text{NCE}$ plot, as they are smaller in magnitude. Since the color map has a large range of values, it is hard to visually distinguish between values in the middle of the plots.
>
> *Regarding Algorithm 1 clarifications:* We apologize for the confusion - the reviewer is indeed correct, and the notation for $\psi$ was overloaded to mean both the instantaneous set of weights, and the weights updated using EMA. We have made the corresponding adjustments to the notation by distinguishing the EMA weights with $\psi_\text{EMA}$. The symbol $\mathcal{B}$ denotes the minibatch of data sampled from the dataset $\mathcal{D}$. The gradients of all losses are estimated using Monte-Carlo samples over the batch of data $\mathcal{B}$.
>
> *Regarding choice of hyperparameters:* The main hyperparameters in all methods were the auxiliary loss coefficients, e.g. BC weight in CVL and CQL, and entropy weight in BC. We have tuned these hyperparameters over the logarithmically-spaced values [0, 0.01, 0.1, 1.0] on a small subset of tasks (drawer and door domains). These are kept constant across all tasks and all experiments (MetaWorld and Mountain Car). We have added these details to the revised manuscript.
>
> *Regarding practical utility of the contrastive value function:* We agree that importance sampling can have high variance; however, our experiments show that our method clearly does have practical utility -- it improves offline RL performance by as much as 210% on some challenging image-based tasks. The high variance of the IS estimator is another reason why using random Fourier features is beneficial during training.
>
> *Regarding clarity of learning an implicit occupancy measure:* We have revised this sentence to clarify that the inputs to the model are the state, action, and future state, and the output is a scalar number indicating whether that future state is likely or not. Please let us know of any other sections that are convoluted, and we will revise them to be more clear.
>
> *Regarding Footnote 3:* We apologize for the confusion. The exact probability mass function of the truncated geometric distribution between M and m is
> $$
> \frac{F_X(y-m)-F_X(y-1-m)}{F_X(M)-F_X(m)}
> $$
> for $F_X$ being the cdf of the geometric distribution. In Equation 12, we denote the geometric sum as the expectation over a geometric random variable over the support $[t,H]$. However, the normalization term $F_X(M)-F_X(m)$ is not present in the original formulation of $Q_\text{NCE}$, hence the proportionality instead of equality on line 2. We could have left the normalization term in the equation, but it does not impact the learning process and hence was left out of the equation.
>
> *Regarding multiple critics:* The argument follows a similar line of reasoning as outlined in Appendix C of [1].
>
> *Regarding experiment duration:* All experiments for MetaWorld were conducted for 1 million gradient steps. This number also includes the pre-training budget for methods that leverage reward-free data (CQL+UDS and CVL with pre-training), which both used 100,000 gradient steps for pre-training.
>
> *Regarding batch size:* The reviewer is correct - the batch data for CVL is **not** i.i.d., as it consists of successive transitions from the same episode, parallelized across multiple devices. For the MetaWorld experiments, where H=500 for all tasks, the CVL batch size is almost identical to the batch size of other methods, which use a batch size of 512.
>
> *Regarding Lemma 1 proof:* The reviewer is correct, we apologize for omitting the expectation and have corrected the typo in the revised manuscript.
>
> *Regarding the introduction of occupancy measure:* As suggested, we have rephrased the first mention of occupancy measure to connect it directly to long-horizon transition models.
>
> [1] https://arxiv.org/abs/2206.07568

---

### Official Review · Reviewer_4mmq · 2022-10-25

**Confidence:** 2
**Correctness:** 4
**Technical Novelty And Significance:** 3
**Empirical Novelty And Significance:** 4
**Recommendation:** 6

**Clarity, Quality, Novelty And Reproducibility:**

This paper seems novel and its technical contributions and validation are high quality. I only have minor concerns about the reproducibility given the many moving parts as well its distinction from several existing paradigms.

Clarification:
- For the experiment in Figure 4, is the no RFF comparison using the full Gaussian kernel (Eq. 12)?

**Strength And Weaknesses:**

Stengths:
- The paper is well motivated, with a detailed background on NCE and its drawbacks (mainly computational), leading to the constrastive loss proposed in Eq.11. The use of the approximate linearized kernel seems novel in its application to this setting.
- The work presents an interesting line of research for value-based offline RL and seems to work well. Rather than improve offline RL with model-based generative models which require expensive rollouts and single-step compounding errors, this work aims to bypass this with a "classification" approach that has worked well in other domains. The empirical validation against other offline benchmarks are convincing.

Weaknesses:
- Figure 5 is a confusing result to showcase what I think is an important detail of this approach. Is there a quantitative measure of the similarities between distributions, e.g. MMD. I agree that this is difficult given one only cares about the topology and not the actual scores, as the former is enough for Q-estimates to be useful for RL, but I am not convinced by this result as is.
- I have some concerns about the robustness of this approach with generalization to out of unseen examples, i.e. extrapolation error. In particular, it does not seem compatible with model-based offline RL methods, e.g. COMBO, which aim to mitigate this issue with a learned dynamics model. Could the authors provide any insight into how their C-learning method trades off the "extrapolation vs interpolation" as seen in offline RL.
- Along the lines of the above, it would be interesting to see an ablation study of the method's learned policy with and without the behvior cloning term in Eq. 15

**Summary Of The Paper:**

This paper proposes a novel contrastive learning algorithm to estimate the occupancy measure of future states in the offline RL setting. It tackles the well-known issue of extrapolation error in existing algorithms for offline RL, in which rewards outside of the underlying data distribution is not well captured by the model. The contrastive method presented is also able to predict long-term state occupancies without having to rely on an autoregressive model. Additionally, the implicit model captures the full distribution of the occupancy measures rather than just the first moments as seen in prior work.


**Summary Of The Review:**

Overall, I think this is a nice paper and would of interest to the wider RL community for its theoretical and practical algorithmic contributions. I would lean towards accepting it.
-------------
Edit: I thank the authors for their response. Although they addressed my concerns, after discussion, I concur with other reviewers about the issues around the clarity of the technical details. Moreover, I believe there is merit in the work and its improved exposition could make it a strong piece of work. In particular, I encourage the authors to detail how the work differs from and connect back to Eysenbach et al. 2022.

---

> ### Author Response · Authors · 2022-11-16
> **Response to reviewer 4mmq**
>
> Dear reviewer,
>
> We thank the reviewer for the informative feedback - we agree with most of the comments and provide a detailed response below. Please let us know of any additional questions or concerns arise.
>
> *Regarding quantitative measures of similarities between distributions:* Thank you for bringing this to our attention, we realize that we have only provided a qualitative analysis of the similarity between TD and contrastive Q-values. We added additional quantitative metrics to Sec. 6.3.2, which show how the MMD decreases over training gradient steps.
>
> *Extrapolation vs interpolation trade-off:*  The main mechanism for handling extrapolation is the policy regularization term, introduced in Eq. 15. When we use a large regularization coefficient, the policy avoids sampling OOD actions, even ones that have high predicted Q-values. Our experiments demonstrate that a 0.1 value of this regularization coefficient works well in practice, and results in a method that can outperform strong baselines (e.g., CQL+UDS).
>
> *Ablation on the BC term:*  As suggested by the reviewer, we ran an additional ablation experiment to study the effect of this BC term. The results, shown in the new Figure 7, show that while adding a behavior cloning regularization term improves the performance of CVL, this difference is marginal and does not account for the major improvement of CVL over BC-only or CQL-only results.
>
> *Linearized vs full Gaussian kernel ablation:* Figure 4 already includes the comparison of the linearized kernel (with RFF) to the full Gaussian kernel (no RFF). Using the full Gaussian kernel in fact leads to lower performance than the RFF approximation: the linearized kernel allows to keep track of reward-weighted future features using EMA, therefore lowering the variance of the gradient estimator and stabilizing the learning process.

---

### Official Review · Reviewer_WeYu · 2022-10-26

**Confidence:** 3
**Clarity, Quality, Novelty And Reproducibility:** Some parts of the paper are very vagu…
**Correctness:** 2
**Technical Novelty And Significance:** 3
**Empirical Novelty And Significance:** 3
**Recommendation:** 5

**Strength And Weaknesses:**

Strength
- The paper proposes a new RL algorithm which uses contrastive learning to learn an implicit dynamics model. The implicit model is learned to capture the occupancy measure and is then used to estimate a contrastive Q-function which is proportional to the actual Q-function.

- The contrastive Q-function is verified in the numerical experiments to have a similar shape as the SAC Q-function in the continuous Mountain Car environment.

- In the numerical experiments of MetaWorld benchmarks show good performance of the proposed CVL algorithm.

Weaknesses
- It is not clear from the paper how positive and negative samples are generated for contrastive learning. In (7), positive and negative samples are both denoted by $\Delta t$ and the distributions for the samples are not specified. In (11) the negative samples are stated to be generated by a truncated geometric distribution, but there is no description on how the parameter $t'$ is selected. Note that this part is different from Eysenbach et al. (2022) where the positive samples are generated by the occupancy measure and negative samples are random state-action pairs.

- Although occupancy measure does not depend on time, what is learned by contrastive learning seems to be the log likelihood ratio of the $\Delta t$-ahead state given by the RHS of (8). This log likelihood ratio in the RHS of (8) is not the occupancy measure and it should depends on the time difference $\Delta t$. However, the proposed contrastive learning method attempts to learn the $\Delta t$-dependent RHS of (8) by the $\Delta t$-independent LHS of (8). There seems to be something wrong, or I missunderstand some part of the contrastive learning objective.

- The paper claims to focus on offline RL, but the offline aspect is not clearly discussed in the paper. The estimated contrastive Q-function is said to be the Q-function of the policy $\pi$, but since it's learned using the offline dataset, it actually won't correspond to the Q-function of the current policy. It is very confusing which the policy $\pi$ refers to in different parts of the paper.


**Summary Of The Paper:**

The paper proposes a contrastive learning based method for offline RL where a implicit dynamic model is learned based on contrastive learning. A contrastive Q-function is calculated based on the implicit dynamic model and the Q-function is then used in policy updates. Numerical experiments on robotic manipulation tasks show improved performance compared with prior methods.

**Summary Of The Review:**

The proposed new contrastive learning based RL algorithm looks promising, but there are some concerns regarding how the contrastive learning model actually learns.

=== after the rebuttal
From the authors' reply, some of the "random variables" like the $\Delta t$ in the LHS of (8) are actually non-random and are meant to be the averaged values over multiple samples. So those values should be denoted by the corresponding expected values, but which are expected values and which are random variables is very confusing and there are still incorrect steps in the latest version. The paper may contains some good ideas, but a major revision with better clarification seems to be necessary.

---

> ### Author Response · Authors · 2022-11-16
> **Response to reviewer WeYu**
>
> Dear reviewer,
>
> Thank you for the insightful comments - we have made the necessary adjustments in the revised manuscript version, and provide detailed answers below.
>
> *How are positives/negatives sampled?:* We have revised Sec. 4 to clarify how these are sampled. The positive samples come from the discounted state occupancy measure: we first sample a time offset $\Delta t \sim \text{Geometric}(1-\gamma)$ (column in the dataset), and then sample a state from the distribution of states at this given offset (row in the dataset). As per classical InfoNCE formulation, this form the joint distribution $(s_t, a_t, s_{t+\Delta t})$, which is contrasted against the negative distribution of product of marginals $p(s_t, a_t) \times p(s_{t+\Delta t})$.
>
> *Time dependence in Eq.8:* We have revised Section 3 to clarify that the proposed method uses contrastive learning to estimate a time-independent quantity. In Eq.8, $S_{t+\Delta}$ is a random variable quantity, meaning that the RHS is a function of a random variable. The way that we remove the explicit dependence of the LHS on $\Delta t$ is through averaging over multiple samples of future state indices.
>
> *Q-values depend on $\pi$:* We have revised the notation throughout the paper to use $Q^\mu(s, a)$ to refer to Q-values corresponding to the behavioral policy.. We note that prior offline RL methods also use $Q^\mu(s, a)$ to drive the policy improvement step. Indeed, using $Q^\mu(s, a)$ for the policy improvement step still guarantees that the learned policy achieves higher returns than the behavioral policy, $\mu(a \mid s)$ [Kakade and Langford].

---

### Decision · Program_Chairs · 2023-01-20

**Decision:**

Reject

**Justification For Why Not Higher Score:**

The clarity issues for this paper were a deal-breaker.  Key aspects of the algorithm were left underspecified.


**Justification For Why Not Lower Score:**

n/a

**Metareview: Summary, Strengths And Weaknesses:**

(a) Summary: The paper proposes a new algorithm that learns an estimate of the distribution of future state occupancy (for a random distribution of future times) conditional on the current state and action; these implicit dynamics are used to estimate a $Q$-function, which is used for policy updates.  This approach aims to avoid having to learn an explicit model of dynamics, while still accounting for changes in distribution between the behavior policy and the learned policy.

(b) Strengths: The idea seems like a promising direction; the experiments provide some evidence of its efficacy.

(c) Weaknesses: The paper does not describe its algorithm with sufficient clarity; many crucial details are missing.  It's good that the authors plan to release code for the algorithm, but the paper should describe it in enough detail to make the main points clear without the code.

**Summary Of Ac-Reviewer Meeting:**

Reviewer 4mmq, Reviewer qU1j, Reviewer WeYu, and I met on November 30 to discuss the paper.

All reviewers agreed that the presentation needs a lot of work.  Reviewer qU1j considers the clarity issues a deal-breaker; it's not clear to him where the performance improvements are coming from.  Reviewer WeYu has nearly identical concerns; he's not sure how he would implement the paper's algorithm based only on its description.  It isn't clear to either reviewer what this paper used for its negative and positive examples.

Reviewer 4mmq was the most positive reviewer.  He acknowledged that a lot of his positivity came from "reading between the lines" of the paper.  He has worked in this area, and in his experience getting a new algorithm to work at all is a major achievement.

In the end we agreed to recommend rejection; there may be a good idea here, but the paper needs to do a better job of describing its contribution and evaluating its results.